# Correlations between Environmental Factors and Milk Production of Holstein Cows

**Roman Mylostyvyi** * **and Olexandr Chernenko**

Department of Biotechnology and Animal Health, Dnipro State Agrarian and Economics University, Dnipro 49100, Ukraine
* Correspondence: mylostyvyi.r.v@dsau.dp.ua

**Abstract:** Global climate change is a challenge for dairy farming. In this regard, identifying reliable correlations between environmental parameters and animals' physiological responses is a starting point for the mathematical modeling of their effects on the future welfare and milk production of cows. The aim of the study was to examine the relationship between environmental parameters and the milk production of cows in hot period. Archival data from the Ukrainian Hydrometeorological Center were used to study the state of insolation conditions (IC), wind direction (WD), wind strength (WS), air temperature (AT), and relative humidity (RH). The temperature–humidity index (THI) (Kibler, 1964) and temperature–humidity index in the hangar-type cowshed ($THI_{CHT}$) (Mylostyvyi et al., 2019) served as integral indicators of the state of the cowshed's microclimate. The daily milk yield (DMY), yield of milk fat (MF) and milk protein (MP), and percentage of milk fat (PMF) and protein (PMP) were taken into account by the DairyComp 305 herd management system (VAS, USA). Statistical data processing was performed using the mathematical functions of Microsoft Excel (Microsoft Inc.) and Statistica 10 (StatSoft Inc.). There was a weak correlation between IC and DMY at r = −0.2, between RH and DMY at r = +0.4, and between RH and MF at r = +0.2. Between DMY, MF, MP, and WS made up r = −0.2 to 0.4. Between DMY, MF, MP, and AT made up r = −0.2 to 0.5 ($p < 0.05$). The effects of weather factors on animal productivity will be the subject of further research.

**Dataset:** Available in Supplementary Materials.

**Keywords:** weather conditions; cows; milk yield; milk fat and protein; correlation

---

## 1. Introduction

High air temperatures adversely affect the welfare and productivity of dairy cows during the summer, not only in tropical regions, but on most of the European continent. This is the cause of significant losses of livestock [1,2]. However, high temperature should not be considered the only factor leading to decreased cow milk yield in the hot period of the year. Humidity, air velocity, and insolation can significantly affect animal physiology, acting together on their thermoregulation [3,4]. Special indices are used for comprehensive assessment of the effects of these environmental factors on animals. Of these, the most common is the temperature–humidity index (THI), which takes into account the effects of air temperature and relative humidity on animals [5–7]. Indices that take into account the mobility of air, the intensity of insolation, and other factors besides temperature and humidity were also proposed not so long ago [8–10]. As a rule, they are all built taking into account the close connection (correlation) between environmental factors and individual indicators of the functional state of cows, which act as specific predictors [11,12]. Milk yield and milk components can

also be indicators of animal comfort, since they are affected by the physiological state of the organism subject to high temperature [13]. Many researchers have taken into account data of meteorological stations located near farms when assessing the impact of weather on animal productivity [14,15]. Such an approach is acceptable when evaluating the effect of heat on animals, not only during grazing but also in uninsulated barns in which the microclimate is similar to the state of the environment [16]. Despite the general availability of weather data (for example, on national weather sites), they contain a large array of records and their systematization before statistical computer processing takes a lot of time. Private information (in our case, the productivity of cows in a separate dairy complex) can be secret and inaccessible in most cases, which limits its use (verification) by other researchers. In this regard, the aim of the work was to study the correlation between environmental parameters and milk production indicators of Holstein cows during the warm period of the year. The results and data presented in this paper will be used for mathematical modeling of the influence of weather conditions on the milk productivity of cows during periods of heat in our future studies.

## 2. Data Description

Data on weather conditions and milk productivity of cows were collected during the warm season. The analysis was carried out on the basis of weather data provided by the Ukrainian Hydrometeorological Center in the warm months of the year (May through August 2017); they are publicly available on the Internet at Meteo.ua (https://meteo.ua/) in the weather archive section. The data include 2912 records (in the Supplementary Materials Data_1.xls file), and log indicators such as insolation conditions (IC), wind direction (WD), wind strength (WS), air temperature (AT), and relative humidity (RH). Temperature–humidity indices (THI and $THI_{CHT}$) were calculated by taking into account AT and RH using the appropriate formulas (described in the Methods section). These data have been added to the Supplementary Materials Data_2.xls file. The average values (for each day) for all indicators were calculated using the built-in mathematical formulas in Microsoft Excel for the four warm months. Accordingly, there were 123 entries for each weather indicator (IC, WD, WS, AT, RH, THI, and $THI_{CHT}$). These data are given in the Supplementary Materials Data_3.xls file.

Data on the productivity of cows included daily milk yield (DMY), milk fat (MF) and milk protein (MP), and percentage of milk fat (PMF) and protein (PMP), located in the Supplementary Materials Data_4.xls file.

Averages were taken for mathematical processing of all indicators (in the Supplementary Materials Data_5.xls file). These data contain 123 complete records of the mean values that were taken for the correlation analysis in this study.

## 3. Methods

The dairy complex is located in an open area (48°28′44″ N, 35°36′46″ E) near the city of Pavlograd. Cows are kept without restraint on the dairy complex. This large dairy complex is designed for many milk cows. All the animals are divided into technological groups (early, middle, and late lactation). The study was conducted on cows in middle lactation (91 to 210 days). The average number of months ranged from 748.4 to 772.5. The uninsulated barns, in the form of hangars, are equipped with canvas curtains that remain open during warm periods. Animals rest in four rows of cubicles with an area of 2.24 $m^2$ per cow. The area per cow in the barn is 3.4 $m^2$ (without a feeding alley). Sand is used as bedding. The cows are fed the same type of nutrient-balanced feed mixture all year round. The feed rationing is made up of silage and haylage from corn, straw from winter and spring crops, and concentrates. The composition and energy value of food depends on the physiological state and productivity of the cows. The animals have free access to water (group drinkers for animals) and feed (table for feeding). The climate features in the barn we described previously [17] are similar to the state of the environment. Large-diameter fans work indoors around the clock during the warm period. The average air velocity is 0.5–0.9 m/s at the animal resting place. Since the feeding of animals is the same throughout the year, this factor can be leveled to a certain extent.

The distance between the cowsheds and the weather station located in Pavlograd city does not exceed a straight 25 km. Geographically, this area belongs to the steppe of Ukraine, which is characterized by relatively constant weather conditions over a large area. Therefore, we considered the data taken from a nearby meteorological station to be quite relevant. Systematization of various weather data was carried out using a key (codes). Insolation conditions (IC) were characterized as follows: clear weather: 4; overcast prevails: 3; cloudy and raining: 2; cloudy constant: 1. This rating from the brightest illumination (4 points) to less bright (1 point) was chosen due to the lack of a quantitative characteristic of the intensity of solar radiation (in $W/m^2$ or derived units) at the weather station. The wind was considered according to cardinal direction: northern: 1; northeastern: 2; eastern: 3; southeastern: 4; southern: 5; southwestern: 6; western: 7; northwestern: 8. Such coding may not have been entirely correct from a logical point of view (since the average data obtained could not be quantified), but we used this for mathematical data processing anyway. Wind speed (strength) was quantified by points [18]: 0–0.5 m/s: 0; 0.6–1.7 m/s: 1; 1.8–3.3 m/s: 2; 3.4–5.2 m/s: 3; 5.3–7.4 m/s: 4; 7.5–9.8 m/s: 5; 12.5–18.2 m/s: 6; ≥18.3 m/s: 7. If a necessary criterion with which to characterize wind strength (for example, wind speed of 10 m/s) was absent, it was assigned to the nearest value (in this example, 5). The cowsheds are located from north to south in relation to the Earth's directions. In the warm season, the curtains of the uninsulated cowsheds are left lowered, which could additionally affect the air exchange in the rooms despite the operating fans. Therefore, we took into account the wind speed and direction outside the building. Air temperature (°C) and relative humidity (%) were recorded in corresponding values. Initially, the weather data was supposed to be recorded every hour to calculate the average values per day (if they were in the weather archive). Calculations of the temperature–humidity index and temperature–humidity index in the cowshed hangar were made according to the following equations:

$$THI = 1.8 \times T - (1 - RH/100) \times (T - 14.3) + 32 \qquad (1)$$

$$THI_{CHT} = 46.00549 + 1.04460 \times T \qquad (2)$$

where THI is temperature–humidity index [19], $THI_{CHT}$ is temperature–humidity index in the hangar-type cowshed, T is ambient air temperature (°C), and RH is relative humidity (%). $THI_{CHT}$ is an indicator we suggested previously [17], on the basis of repeated temperature and humidity measurements in an animal room throughout the year. This indicator was used because the animals in the study were kept in exactly a hangar-type barn. $THI_{CHT}$ allows determination of the temperature–humidity index for only one indicator (external temperature), taking into account the design features of the cowshed. Data on cow productivity (milk yield per herd, number of dairy cows, daily milk yield, milk fat and protein yield, as well as fat percentage and protein in milk) were obtained in the conditions of the dairy complex by the DairyComp 305 herd management system (VAS, USA). The relationship between the state of the environment and the productivity of dairy cows was assessed using Statistica 10 software (StatSoft Inc.). Correlation analysis was done using Spearman's rank correlation coefficient. Significance was determined to be $p < 0.05$.

## 4. Results

### 4.1. Weather Conditions

The weather conditions were studied from 1 May to 31 August 2017. We estimated the weather for a duration of 2912 h, which is about 98.6% of the warm period of the year, lasting 123 days (2952 h). It was found (Table 1) that there were 2255 h with clear weather (77.4%), it was mostly cloudy for 527 h (18.1%), it was rainy for 95 h (3.3%), and continuous cloudiness lasted 35 h (1.2%). In May, it was cloudy more often; in July, it was rainier; and the number of clear days was greatest in August.

**Table 1.** Characteristics of weather conditions.

| Month | Clear Weather | | Overcast Prevails | | Cloudy and Raining | | Cloudy Constant | |
|---|---|---|---|---|---|---|---|---|
| | Hours | % | Hours | % | Hours | % | Hours | % |
| May | 529 | 71.8 | 165 | 22.4 | 27 | 3.6 | 16 | 2.2 |
| June | 563 | 78.7 | 134 | 18.7 | 18 | 2.6 | – | – |
| July | 534 | 73.7 | 138 | 19.0 | 45 | 6.2 | 8 | 1.1 |
| August | 629 | 85.6 | 90 | 12.2 | 5 | 0.7 | 11 | 1.5 |

It was established (Figure 1) that the prevailing winds were from the north (30.9%), the west (16.8%), the east (15.2%), and the northeast (15.0%) in the warm season. The southeast wind direction was not fixed in one of the cases.

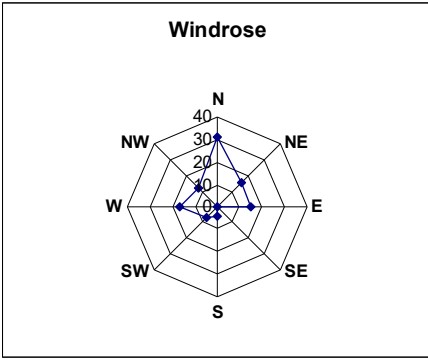

**Figure 1.** Directions of prevailing winds during the study period.

As shown by month in a separate table (Table 2), the prevailing winds were from the north in May, June, and July. In August there was more wind from the east (the north and northeast also had a fairly large share).

**Table 2.** Direction of the wind.

| Wind Direction | May | | June | | July | | August | |
|---|---|---|---|---|---|---|---|---|
| | Hours | % | Hours | % | Hours | % | Hours | % |
| Northern | 233 | 31.6 | 210 | 29.4 | 275 | 37.9 | 182 | 24.8 |
| Northeastern | 148 | 20.1 | 53 | 7.4 | 72 | 9.9 | 165 | 22.4 |
| Eastern | 92 | 12.5 | 63 | 8.8 | 70 | 9.7 | 217 | 29.5 |
| Southeastern | – | – | – | – | – | – | – | – |
| Southern | 41 | 5.6 | 39 | 5.4 | 35 | 4.8 | – | – |
| Southwestern | 68 | 9.2 | 65 | 9.1 | 38 | 5.2 | 12 | 1.6 |
| Western | 104 | 14.1 | 147 | 20.6 | 133 | 18.4 | 104 | 14.2 |
| Northwestern | 51 | 6.9 | 138 | 19.3 | 102 | 14.1 | 55 | 7.5 |

The total amount of time that the wind strength (Table 3) was equal to 0 points was 221 h (7.6%); 1 point, 97 h (3.3%); 2 points, 1022 h (35.1%); 3 points, 956 h (32.8%); 4 points, 494 h (17.0%); and 5 points, 122 h (4.2%). Wind strength of 6 and 7 points was not recorded.

**Table 3.** Wind strength.

| Points | May | | June | | July | | August | |
|---|---|---|---|---|---|---|---|---|
| | **Hours** | **%** | **Hours** | **%** | **Hours** | **%** | **Hours** | **%** |
| 0 | 56 | 7.6 | 45 | 6.3 | 81 | 11.2 | 39 | 5.3 |
| 1 | 34 | 4.6 | 29 | 4.0 | 26 | 3.6 | 8 | 1.1 |
| 2 | 269 | 36.5 | 278 | 38.9 | 294 | 40.6 | 181 | 24.6 |
| 3 | 233 | 31.6 | 238 | 33.3 | 209 | 28.8 | 276 | 37.6 |
| 4 | 111 | 15.1 | 102 | 14.3 | 98 | 13.5 | 183 | 24.9 |
| 5 | 34 | 4.6 | 23 | 3.2 | 17 | 2.3 | 48 | 6.5 |

The windiest time was in August, when wind force from 3 to 5 points was 507 h, or 69% of the total time. However, the duration of "calm" was also longest this month, at 5.3%. Average values of air temperature and relative humidity are given as background indicators in the warm period of the year (Table 4).

**Table 4.** Average temperature and relative humidity condition of air, mean ± standard deviation (SD). THI, temperature–humidity index; $THI_{CHT}$, temperature–humidity index in hangar-type cowshed.

| Month | | Air Temperature | | Relative Humidity | | THI | | $THI_{CHT}$ | |
|---|---|---|---|---|---|---|---|---|---|
| | **n** | **Mean** | **SD** | **Mean** | **SD** | **Mean** | **SD** | **Mean** | **SD** |
| May | 31 | 15.6 | 0.64 | 59.9 | 2.43 | 58.8 | 0.87 | 62.3 | 0.67 |
| June | 30 | 20.9 | 0.59 | 58.3 | 1.79 | 66.3 | 0.80 | 67.8 | 0.62 |
| July | 31 | 21.6 | 0.62 | 63.9 | 1.83 | 67.4 | 0.79 | 68.5 | 0.65 |
| August | 31 | 24.2 | 0.82 | 50.6 | 2.66 | 69.9 | 1.00 | 71.5 | 0.86 |

However, the average values of the indicators of air temperature and relative humidity, as well as indices calculated taking them into account, do not reflect the duration of the hot period. Accordingly, the distribution of their values in time is given more specifically (Table 5).

**Table 5.** Distribution of temperature and humidity index values (THI and $THI_{CHT}$) over time in the warm months of the year (h).

| Month | THI | | | | $THI_{CHT}$ | | | |
|---|---|---|---|---|---|---|---|---|
| | **<68** | **68.0–71.9** | **72.0–79.9** | **80.0–89.9** | **<68** | **68.0–71.9** | **72.0–79.9** | **80.0–89.9** |
| May | 649 | 73 | 15 | – | 606 | 87 | 44 | – |
| June | 429 | 152 | 133 | 1 | 418 | 123 | 168 | 6 |
| July | 386 | 146 | 188 | 5 | 391 | 131 | 184 | 19 |
| August | 288 | 148 | 245 | 54 | 253 | 156 | 206 | 120 |

Note: The gradation of temperature and humidity index values was made as follows: values below 68 correspond to comfortable conditions for cows, 68–71 corresponds to slight stress, 72–79 to moderate stress, and 80–89 to strong stress [17].

These data show that THI and $THI_{CHT}$ values were high enough for dairy cows to be uncomfortable [20], especially from June to August when the values increased tenfold.

*4.2. Milk Production of Cows*

Analysis of the milk production of cows was conducted from 1 May to 31 August 2017. The average daily milk yield and the content of its main components were taken into account for the herd (Table 6).

**Table 6.** Milk productivity of cows, mean ± SD.

| Month | DMY (kg) | | MF (kg) | | MP (kg) | | PMF (%) | | PMP (%) | |
|---|---|---|---|---|---|---|---|---|---|---|
| | Mean | SD | Mean | SD | Mean | SD | Mean | SD | Mean | SD |
| May | 23.7 | 0.03 | 0.851 | 0.005 | 0.765 | 0.004 | 3.60 | 0.020 | 3.23 | 0.018 |
| June | 23.8 | 0.02 | 0.834 | 0.004 | 0.756 | 0.005 | 3.50 | 0.014 | 3.17 | 0.020 |
| July | 24.0 | 0.05 | 0.833 | 0.003 | 0.746 | 0.006 | 3.47 | 0.010 | 3.11 | 0.023 |
| August | 23.3 | 0.05 | 0.804 | 0.005 | 0.725 | 0.006 | 3.45 | 0.018 | 3.11 | 0.022 |

DMY, daily milk yield; MF, yield of milk fat; MP yield of milk protein; PMF, percentage of milk fat; PMP percentage of milk protein.

We assumed that May was the most comfortable for cows in terms of weather conditions. In the summer months, there was a noticeable decrease in the components of milk compared with May. We observed a decrease in milk fat yield by 17–47 g, milk protein yield by 9–40 g, milk fat content by 0.1–0.15%, and milk protein by 0.06–0.12%. Differences between May and July–August were significant ($p < 0.05$–0.001). We attribute such changes in the milk composition to the influence of environmental factors on animals, because all the cows had the same lactation period.

*4.3. Correlations between the State of the Environment and Milk Production Indicators*

Considering the relationship between environmental factors and the productive qualities of cows (Table 7), we noted a reliable correlation. The correlation was positive between IC and DMY (r = –0.2), as well as between RH and DMY (r = +0.4) and between RH and MF (r = +0.2). A negative correlation of different densities was obtained between DMY, MF, MP, and WS (r = –0.2 to 0.4). The milk production indicators correlated negatively with AT (r = –0.2 to 0.5) in all cases, and, for $THI_{CHT}$, the correlation between the productivity of cows was significantly negative with a similar strength.

**Table 7.** Correlations (r) between the state of the environment and milk production of cows.

| State of the Environment | DMY | MF | MP | PMF | PMP |
|---|---|---|---|---|---|
| IC | −0.195 * | −0.168 | −0.110 | −0.095 | −0.046 |
| WD | +0.090 | +0.015 | −0.006 | −0.023 | −0.035 |
| WS | −0.409 * | −0.206 * | −0.189 * | −0.046 | −0.059 |
| AT | −0.186 * | −0.465 * | −0.366 * | −0.443 * | −0.333 * |
| RH | +0.399 * | +0.225 * | +0.126 | +0.066 | −0.009 |
| THI | −0.113 | −0.447 * | −0.354 * | −0.457 * | −0.345 * |
| $THI_{CHT}$ | −0.187 * | −0.466 * | −0.367 * | −0.444 * | −0.333 * |

* Reliable values are shown ($p < 0.05$).

Thus, weather conditions such as the intensity of solar radiation, wind strength, air temperature, and humidity should be taken into account when assessing the influence of the environment on the milk productivity of cows.

## 5. Conclusions

This paper demonstrates the possibility of using meteorological data available to a wide range of people in order to identify the relationship between the state of the environment and the productive qualities of dairy cows, as a specific example. Some approaches to summarizing individual meteorological data before conducting a correlation analysis will be useful for researchers working in toward this end. The dataset presented here can be reproduced by other researchers, which will serve as a basis for new ideas and interpretations.

## 6. User Notes

The key (codes) listed in the sections above will allow researchers to reproduce the results obtained in the paper (using Excel files in the public domain).

**Supplementary Materials:** The following are available online at http://www.mdpi.com/2306-5729/4/3/103/s1, Text S1: Weather and cow productivity data.

**Author Contributions:** R.M. performed methodology, software validation, and original draft preparation; O.C. performed writing—review and editing.

**Funding:** This research received no external funding.

**Acknowledgments:** The authors are grateful to Halyna Novokshonova for the materials submitted for the study, as well as to students of the biotechnology faculty, Anastasia Sokolan and Alyna Popovych, for collecting primary data.

**Conflicts of Interest:** The authors declare no conflict of interest. The funders had no role in the design of the study; in the collection, analysis, or interpretation of data; in the writing of the manuscript, or in the decision to publish the results.

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
