# Peer review of "Correlations between Environmental Factors and Milk Production of Holstein Cows"

_data_

Round 1
Reviewer 1 Report
This is a topical area of research and it is important to understand how we can mine data to predict the likelihood of heat events and cow discomfort. I am not really sure about the format for the journal data but in other journals I would have expected some discussion of the findings. There is no real discussion in There is not real discussion of the findings. Also, there needs to be an extensive review of the grammar. I have picked up some errors in grammar but have largely focussed on the technical aspects of the manuscript.
Specific comments:
Title (and elsewhere) – this should be correlations not correlation as you are looking at more than one correlation.
Line 53 – 58 – This reads very clumsily. Please reword.
Line 86 – five thousand dairy cows.
Line 88 – not sure what “Their average number for months was 748.4 to 772.5 goals.”
Line 88 – The uninsulated barns….
Line 102 – The authors need to justify that weather stations 25 km away represent the weather at the farm. This is a considerable distance and microclimates can exist. Do they have any supporting data apart from the statement that “the area is characterized by a relative constancy of weather conditions over a large area”?
Line 129 – animals were taken into account
Line 142 – Table 1 (here and elsewhere).
Line 144 to 145 – poorly worded sentence
Line 147 – from the north?
Line 151 to 152 – poorly worded sentence
Line 173 to 174 – need some references to support this statement.
Line 190 – what is the definition of a “reliable correlation”?
Line 195 – what is a similar density when discussing correlations?
Author Response
Dear Reviewer. We make explanations for the comments made.
The average number of cows corresponds to the data by months, they are taken from the Data_4 file. We did not compare the meteorological indicators near the premises with the data provided by the weather station in this study (we will take this into account in our work in the future). The number of milk cows and their milk yield were taken into account daily during the specified period of research. “Reliable correlation” is the one that had significant values (P <0.05). By density was meant the strength of correlation (this will be corrected). Some other edits made in the manuscript will be highlighted in a different color.
Your technical expertise of the manuscript deserves deep respect. We will entrust all editing of the grammar to MDPI Author Services, as you recommend us to do.
Regards, authors
Reviewer 2 Report
In general article is starting point for future investigation on milk productivity and environmental factors.
I not fully understand the the conditions were cows are kept. The feed was "artificial", it meand prepared of supplied to the cows. No natural grass was available, so only parameter, defining cow milk production, was environment (temperature etc). But cows were kept in hangars, open during summer, close during winter, and authors apply wind direction analysis for the cows kept in hangar. It looks that parameters such wind and cows behavior can't be used simultaniuosly due hangar effect.
Anyway I wish to authors continue investigate cow productivity, and use more sophisticated parameters.
Author Response
Dear Reviewer. Thank you for your careful attention to our manuscript. All your wishes deserve attention and will be taken into account in our research in the future.
Regards, authors
This manuscript is a resubmission of an earlier submission. The following is a list of the peer review reports and author responses from that submission.
Round 1
Reviewer 1 Report
Dear Autors
The Article is appropriately structured. However, I still recommend that the some concepts should be amended and clarified specified below in order to strengthen it.
L. 9-11 I do not agree with this statement. The starting point for building a mathematical model is the correlation between environmental parameters and the physiological reaction of cows. The sentence needs to be changed.
L. 11 The aim of the study was to study the correlation … The sentence needs to be better structured.
L. 12 Not weather but selected parameters of weather.
L.43-47. I propose to add Herbut P., Angrecka S., Walczak J. 2018. Environmental parameters to assessing of heat stress in dairy cattle – a review. International Journal of Biometeorology. 62(12); 2089–2097
L. 86 Intensity of solar radiation (ISR) … The intensity of solar radiation is the amount of solar power per unit area. Rather it would be better to use e.g the expression insolation conditions (IC) or similar. After the change, it applies to the whole article
L. 101 Air temperature (degrees Celsius) … use (oC)
L.148 In Table 5, the time unit in the title of the table or table is missing (hour - h).
L.161 It needs to be clarified and improved. Not cattle stalls but pens or cubicles with an area of 2.4 m2. Please specify how many m2 per cow in the barn (without a feeding alley).
L. 182-195 I suggest you remove it. This automatism of correlation introduces many doubts that have not been clarified. It also does not contribute anything significant to the content of the article. Correlation at the level of -0.243 is a weak correlation and not a significant one.
Reviewer 2 Report
The fundamental comments are the following:
1. There is no chapter Introduction.
2. Information about 'Terms of keeping cows' should be included in the chapter Material and methods.
3. A too small number of animals in the experiment.
4. The distance between the cowsheds and the weather station (about 25 km) was too far, considering the variability of meteorological conditions.
5. There is no description of statistical methods.
6. There is no chapter Discussion.
7. The Conclusion is inadequate to the results obtained (correlations between meteorological parameters and milk performance of cows were too weak).